# Classification and Morphometric Features of Pterion in Thai Population with Potential Sex Prediction

**DOI:** 10.3390/medicina57111282

**Published:** 2021-11-21

**Authors:** Nongnut Uabundit, Arada Chaiyamoon, Sitthichai Iamsaard, Laphatrada Yurasakpong, Chanin Nantasenamat, Athikhun Suwannakhan, Nichapa Phunchago

**Affiliations:** 1Department of Anatomy, Faculty of Medicine, Khon Kaen University, Khon Kaen 40002, Thailand; nongua@kku.ac.th (N.U.); aradch@kku.ac.th (A.C.); sittia@kku.ac.th (S.I.); 2In Silico and Clinical Anatomy Research Group (iSCAN), Department of Anatomy, Faculty of Science, Mahidol University, Bangkok 10400, Thailand; laphatrada.yur@gmail.com (L.Y.); athikhun.suw@mahidol.edu (A.S.); 3Center of Data Mining and Biomedical Informatics, Faculty of Medical Technology, Mahidol University, Bangkok 10400, Thailand; chanin.nan@mahidol.edu

**Keywords:** pterion, skull, suture, morphometric analysis, anatomical variation, machine learning

## Abstract

*Background and Objectives:* The landmark for neurosurgical approaches to access brain lesion is the pterion. The aim of the present study is to classify and examine the prevalence of all types of pterion variations and perform morphometric measurements from previously defined anthropological landmarks. *Materials and methods:* One-hundred and twenty-four Thai dried skulls were investigated. Classification and morphometric measurement of the pterion was performed. Machine learning models were also used to interpret the morphometric findings with respect to sex and age estimation. *Results:* Spheno-parietal type was the most common type (62.1%), followed by epipteric (11.7%), fronto-temporal (5.2%) and stellate (1.2%). Complete synostosis of the pterion suture was present in 18.5% and was only present in males. While most morphometric measurements were similar between males and females, the distances from the pterion center to the mastoid process and to the external occipital protuberance were longer in males. Random forest algorithm could predict sex with 80.7% accuracy (root mean square error = 0.38) when the pterion morphometric data were provided. Correlational analysis indicated that the distances from the pterion center to the anterior aspect of the frontozygomatic suture and to the zygomatic angle were positively correlated with age, which may serve as basis for age estimation in the future. *Conclusions:* Further studies are needed to explore the use of machine learning in anatomical studies and morphometry-based sex and age estimation. Thorough understanding of the anatomy of the pterion is clinically useful when planning pterional craniotomy, particularly when the position of the pterion may change with age.

## 1. Introduction

The pterion is an important anatomical landmark of the skull where the frontal, temporal, parietal and sphenoid bones are articulated. The pterion is located on the temporal side of the skull around 4 cm above the midpoint of the upper border of the zygomatic arch [1]. The pterion is the thinnest and the weakest spot of the skull. and can be used as an anatomical landmark of the anterior branch of the middle meningeal artery, lateral sulcus of brain and especially Broca’s motor speech area [2]. It is also an important approach point to the sphenoid ridge during optic cancer surgery [3]. Therefore, the position of the pterion is of vital importance in neurosurgery. Variations in the patterning of the pterion and its sutural constituents were noted by Broca [2] and Murphy [4]. The pterion patterns can be classified into four types including spheno-parietal, fronto-temporal, stellate and epipteric [4] (Figure 1). The spheno-parietal pattern is a type of pterion in which there is direct contact between the sphenoid and parietal bones. For the fronto-temporal type, the frontal and temporal bones are in direct contact. The stellate pattern includes all bones including the frontal, parietal, sphenoid and temporal bones. Lastly, the epipteric pattern additionally incorporates the epipteric bone, a small sutural bone of the sphenoid and parietal bones. A recent study demonstrated that in the Thai population the proportion of pterion types was 87.3% spheno-parietal, 4.5% fronto-temporal and 8.2% epipteric [5]. However, the morphometric measurements of the pterion in Thai skulls in relations to previously defined anthropological landmarks have never been investigated. In addition, skulls with complete synostosis of the pterion suture, in other words, when the pterion suture is completely ossified, were typically excluded from investigation [6]. Therefore, the association between pterion sutural synostosis and factors such as age or sex has never been investigated.

Conventionally, once results from anatomical studies are obtained, they are manually interpreted by researchers. As a result, the interpretation of results are subjective and biases are introduced by the researchers themselves, which in turn may threaten the validity of the study. Machine learning, on the other hand, utilizes computer programs to unravel patterns within a dataset. These are then incorporated into algorithms used to look for associations and predict future outcomes. The use of machine learning in evidence-based anatomical research has been recently demonstrated [7], which highlights its potential broader application in anatomical studies in the future. However, the utility of machine learning in classification-based and morphometry-based anatomical studies have never been demonstrated.

The aim of the present study is to classify and examine the prevalence of the pterion variations and perform the morphometric analysis using previously defined anthropological landmarks. Machine learning algorithms were used to study the potential influence of sex and age on morphometric measurements of the pterion.

## 2. Materials and Methods

### 2.1. Anatomical Study and Morphometric Measurements of the Pterion

The dried skulls were obtained from Human Bone Warehouse for Research (UHBWR), Department of Anatomy, Faculty of Medicine, Khon Kaen University. The present study was approved by the Office of The Khon Kaen University Ethics Committee in Human Research (approval number: KKU 660201.2.3/2054). A total of 124 dried skulls (248 sides) from 74 males and 50 females were included. Mean age of bone donor was 65.5 years-old (40 to 94 years). Skulls with pathologies such as porotic hyperostosis [8] were excluded. The pterions on both sides of each skull were classified into four types according to the previously established classification system by Murphy [4], including spheno-parietal type, fronto-temporal type, stellate type and epipteric type (Figure 1A–D). Pterions were classified as synostotic when the pterion suture was completely ossified, equivalent to degree 4 of synostosis [2]. After classification, the skulls were photographed. Finally, the morphometric measurements were carried out by measuring the distance from the center of the pterion to six different locations of the skull [9] including PSFZ (distance from the center of the pterion to the anterior aspect of the frontozygomatic suture), PZAN (distance from the center of the pterion to the zygomatic angle), PZA (distance from the center of the pterion to the zygomatic arch), PH (distance from the center of the pterion to Henle’s spine), PMP (distance from the center of the pterion to the mastoid process of the temporal bone), PI (distance from the center of the pterion to the external occipital protuberance) (Figure 1E). Classification and morphometric measurements were performed by two examiners. Any disagreement between the two examiners was resolved by consensus.

### 2.2. Machine Learning Analysis

Machine learning was performed using Weka, a software developed by University of Waikato, New Zealand [10], to predict the influence of sex and age on the pterions’ measurements. Random forest classifier model was employed for sex prediction. Random forest is a supervised learning algorithm for classification that builds multiple decision trees which are then merged to obtain a stable prediction. Number of iterations was set to 128 [11] with 10-fold cross validation. All attributes, including pterion types and morphometric measurements involving the PSFZ, PZAN, PZA, PH, PMP, PI and H-width of both sides, were evaluated. For age prediction, an unsupervised simple linear regression model was used. This model uses the relationship between the data-points to draw a best-fine line, which is used to predict future values. The attribute “sex” was excluded prior to analysis. The remaining settings were set as default.

### 2.3. Statistical Analysis

Difference in proportion of individuals (by sex and side) with each type of pterion was tested using *z*-test of two proportions. Sex differences were tested using independent *t*-test while side differences were tested using dependent (Student’s) *t*-test. Statistical significance was established at *p* = 0.05.

## 3. Results

### 3.1. Prevalence and Classification of Pterion Types

Prevalence and classification of pterion types are shown in Table 1. All four types of pterions were present (Figure 2). The spheno-parietal type was the most common one (154 sides, 62.1%), followed by epipteric (29 sides, 11.7%), fronto-temporal (13 sides, 5.2%) and stellate (3 sides, 1.2%). In one skull with an epipteric pterion, there were two epipteric bones instead of one on the left side (Figure 1E). Complete synostosis of the pterion suture was observed in 46 sides (18.5%) and was only present in males. Complete synostosis of the pterion suture was not associated with age because the mean age in this group did not differ from that of the remaining skulls (*p* = 0.10). The epipteric type was significantly more common in females. Statistically significant differences were not found for side.

Bilaterality of the pterion types is shown in Table 2. The majority of the skulls showed bilateral symmetry in 83.1% including the complete synostosis of the pterion suture. The bi-spheno-parietal type was the most common (54.4%) followed by bi-epipteric (7.3%), and bi-fronto-temporal (4%). The bi-epipteric type was significantly more prevalent in females. The most common type of asymmetric pterion was sphenoparietal-epipteric type, which was present in 10.5%. The bi-epipteric type was significantly more common in females. Ossification or complete synostosis of the pterion suture was only found bilaterally in 23 skulls and was only present in males.

### 3.2. Morphometric Analysis of Pterion Landmarks

Morphometric measurements of the pterion landmarks are shown in Table 3. Measurements were performed in 100 skulls. One skull belonging to a male donor was excluded because the right side was partially damaged. Twenty-skulls where the pterion sutures were bilaterally synostotic were excluded from morphometric analysis. The PMP distance, or the distance between the pterion center to the mastoid process of the temporal bone, was significantly longer in males. The PI distance, in other words, the distance from the pterion center to the external occipital protuberance, was also significantly longer in males. Significant differences in the morphometric measurements were found for between sides except for the PH distance and H-width (Table 3).

### 3.3. Machine Learning Interpretations and Correlation Analysis

Prior to machine learning analysis, ten skulls were excluded because age was not reported. Random forest classifier was employed for sex prediction (Figure 3). Age as attribute was excluded prior to analysis. Our results demonstrated that random forest could predict sex using the given morphometric data with 80.7% accuracy (root mean square error = 0.38). After 70:30 data split for the training and validation, the accuracy was 82.3% (root mean square error = 0.3548). For age prediction, sex as an attribute was excluded and a linear regression was applied. The results showed that age was possibly correlated with PSFZ, PZAN and PI, which set out the scope of correlation analysis. A linear regression was also applied for potential age prediction; however, the results were poor and unreliable (r = 0.141, root mean square error = 12.43).

Correlation analysis was performed for PSFZ, PZAN and PI as potential predictors of age. Values of the left and the right sides were analyzed independently. We found a significant correlation between PSFZ and age in both left and ride sides (*p* < 0.05) (Figure 4A). Correlation between PZAN and age was only found for the left side (*p* < 0.05) (Figure 4B). No significant correlation was found between PI and age (*p* > 0.05).

## 4. Discussion

One of the most common procedures in neurosurgery is pterional craniotomy. This procedure allows access to tumors and lesions in multiple areas of the brain. This procedure requires understanding of the pterion morphometry and its surface landmark. Also, the pterion is the weakest spot of the skull, overlying the anterior branch of the middle meningeal artery. If ruptured in the event of trauma, it could create a hematoma that could lead to fatal hemorrhage [12]. In addition to its clinical significance, the knowledge of pterion variations is of crucial importance in legal medicine, human identification and anthropology.

Our study demonstrated that the most common type of pterion was the spheno-parietal (62.1%), followed by the epipteric (11.7%), fronto-temporal (5.2%) and stellate (1.2%) types. Our results were consistent with multiple previous studies [4,5,9,13,14,15,16,17,18]. Between-study comparison was thoroughly conducted by Dutt et al. [15] and Vasudha et al. [18]. Pterion type is subject to evolutionary and racial variations. The development of the calvarium is synchronously harmonized with the growth of the brain. This may explain the prevalence of the fronto-temporal type in monkey skulls [19] unlike in humans for which the spheno-parietal is the most common type in order to accommodate the growing brain. Wang et al. [19] stated that the etiology of pterion variations was multifactorial including genetic and environmental factors. These factors may explain why pterion morphology varies considerably across studies. Nevertheless, the methodological quality of previous studies may in part contribute to these differences. For example, it is generally known that the stellate type is the rarest type of pterion. However, a recent study [20] documented a very high prevalence of the stellate type at 25%, although it is worth noting that only 24 skulls were investigated. Overestimation caused by small sample size is known as the small-study effect [21], which reiterates that appropriate sample size is crucial when performing anatomical studies. Another factor which leads to higher heterogeneity between studies is inter-observer variability. For instance, the prevalence of the epipteric type ranges from 0% to up to 51% and, moreover, this type of pterion is not investigated in some studies [15]. Our study found that pterions with of the epipteric type only contained one to two epipteric bones. However, the number of epipteric bones could be up to six [22].

Unlike most previous studies, our study took into account complete synostosis of the pterion suture in our classification. We found that the pterion suture was not identifiable in 46 sides (23 skulls), all of which were found in males. This finding suggests a potential obliteration of the pterion suture. It was previously known that the pterion suture begins obliteration as individuals reach stage 3 from the age of 25 and becomes entirely obliterated at stage 4 after 40 years old [23]. In the present study, it was not possible to detect the early onset of pterion sutural obliteration because the minimum age of our donors was 40 years old with an average of 65.5 years old. It was previously noted that anatomical features are affected by age, and the published anatomical findings are heavily biased towards the elderly population [24]. Our findings also showed that sutural obliteration of the pterion was only observed in males. Although more evidence is needed to better understand the etiology of this phenomenon, it was similarly found that the maxillary suture also underwent obliteration sooner in males [25]. The formation of the neurocranium is a complex process that accommodates the growing brain. Accommodation of the growing brain requires an increase in the cranial volume, which occurs from bone deposition at the osteogenic fronts [26]. Regulatory mechanisms and the interaction between sutures, ossifying bones and the dural reflections orchestrate the whole process [26,27]. Sutures are maintained by an equilibrium between cellular proliferation, migration, differentiation and apoptosis, ensuring a constant balance between growth and separation [28,29]. Sutural obliteration is distinct from craniosynostosis. Craniosynostosis refers to the condition where the cranial sutures prematurely close at an early age. Although the underlying causes of such premature fusion remain unclear [30], it may result in craniofacial deformities and abnormalities [31].

The influence of sex and age on pterion measurements was predicted using machine learning. The application of machine learning in anatomical research was first proposed by Yurasakpong et al. [7], but its utility was only limited to the interpretation of meta-analytic results. In the present study, we piloted the use of machine learning models for sex and age prediction. We found that the accuracy of sex prediction using random forest was about 80% (Figure 3). Although the accuracy was encouraging, especially when pterion measurements have never been used in sex prediction, further improvements can be made to increase it. The composed model could be based on a number of parameters. We found that the PMP and PI distances were significantly longer in males. The longer PMP distance, or distance between the pterion center and mastoid process, could be explained by the fact that the mastoid process and the posterior part of the temporal bone are larger in males than females [32]. We believe the fact that the complete synostosis of the pterion suture was only present in males could in part contribute to the unexpectedly high accuracy. However, it may be problematic as it could lead to overfitting of the model because, although sutural obliteration may be more common in males [25], it is not always sex-specific. In our study, age estimation was performed using a linear regression model. Even though the result was not encouraging because the resulting correlation coefficient was extremely low, it could be used as basis for further analysis. The linear regression results showed that PSFZ, PZAN and PI distances were potential attributes of age; as a result these three parameters were selected for correlational analysis. Using Pearson’s correlation coefficient, we found that the PSFZ and PZAN distances were positively correlated with age (Figure 4A). This finding is in line with the zygomatic bone change that happens with age. It was previously found that the inferolateral aspect of the orbit, which is part of the zygomatic bone, has a tendency to resorb as individuals age, resulting in the lengthening of the lid–cheek junction [33]. This age-related bone change could lead to longer PSFZ and PZAN distances. The present study is not without limitations. Although bone measurements could be associated with height and weight [34], such correlations were not analyzed in this study because height and weight of the donors were unknown.

## 5. Conclusions

The present study investigates the morphometric measurements of the pterion in the Thai population. We found that the pterion suture was obliterated in 18.5%, and it all occurred in males. Although further studies are needed, morphometric measurements of the pterion could potentially be used for sex estimation which may be useful for forensic and anthropological applications. Our study also highlights the potential application of machine learning in anatomical research, especially in terms of age and sex estimation. Finally, precise and accurate localization of the pterion is clinically important when planning pterionial craniotomy, especially when pterion morphometry changes with age.

## Figures and Tables

**Figure 1 medicina-57-01282-f001:**
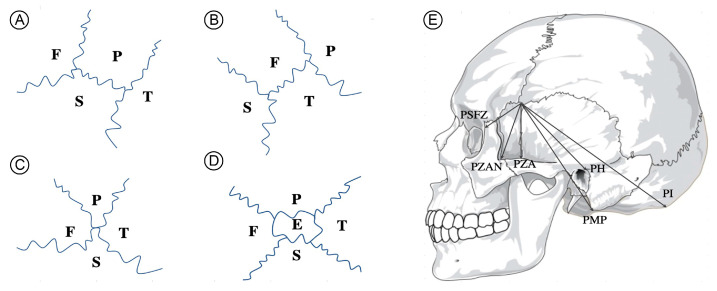
Four types of pterions as described by Murphy, including spheno-parietal (**A**), fronto-temporal (**B**), stellate (**C**) and epipteric (**D**) and morphometric measurements of the pterion (**E**). F—frontal bone, P—parietal bone, T—temporal bone, S—sphenoid bone, E—epipteric bone, PSFZ—distance from the center of the pterion to the anterior aspect of the frontozygomatic suture, PZAN—distance from the center of the pterion to the zygomatic angle, PZA—distance from the center of the pterion to the zygomatic arch, PH—distance from the center of the pterion to Henle’s spine, PMP—distance from the center of the pterion to the mastoid process of the temporal bone, PI—distance from the center of the pterion to the mastoid process of the external occipital protuberance.

**Figure 2 medicina-57-01282-f002:**
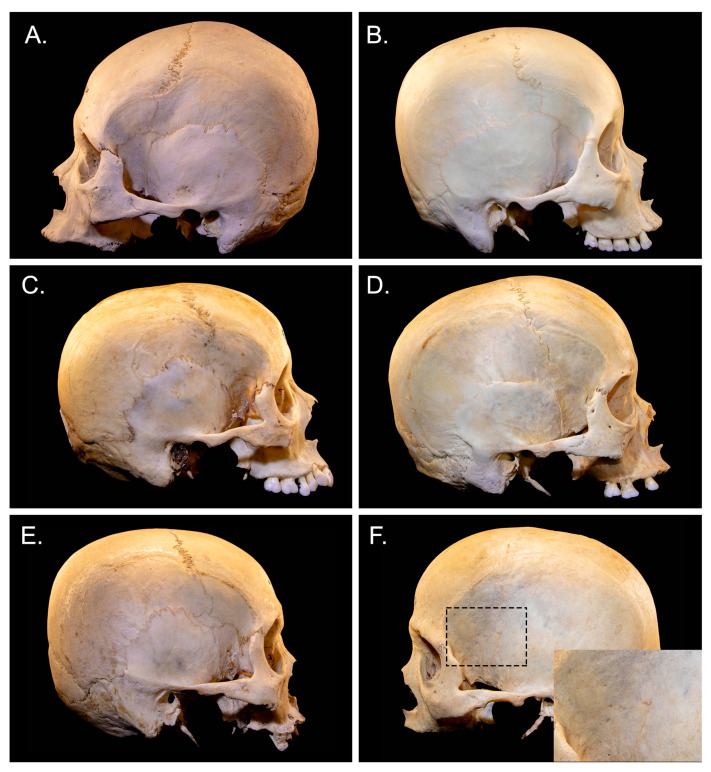
Skeletal images showing morphometric classification of the pterion including spheno-parietal (**A**), fronto-temporal (**B**), stellate (**C**) and epipteric (**D**) types. (**E**) A special type of epipteric pterion with two epipteric bones and (**F**) Complete synostosis of pterion suture with magnification.

**Figure 3 medicina-57-01282-f003:**
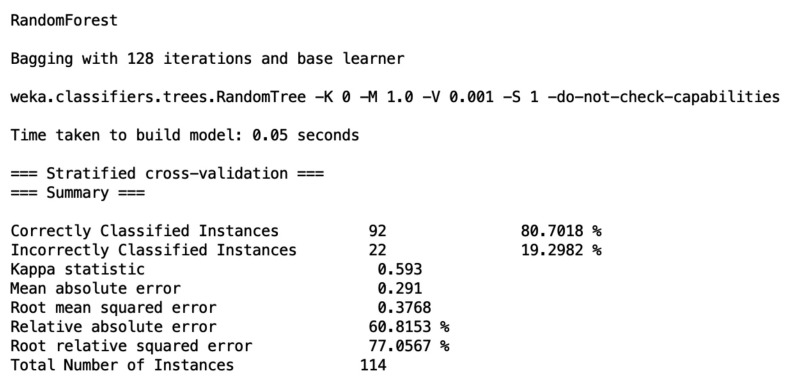
Model output from multivariate analysis on Weka showing random forest model for sex prediction.

**Figure 4 medicina-57-01282-f004:**
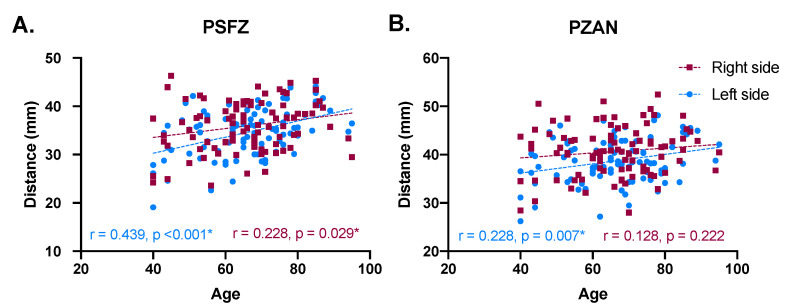
Correlation between age versus PSFZ (**A**) and PZAN (**B**). Correlations were evaluated using Pearson’s correlation coefficient and statistical significance was established at *p* = 0.05.

**Table 1 medicina-57-01282-t001:** Classification of pterion patterns and their prevalence by sex and laterality.

Pterion Types	Total (*n* = 248)	Sex			Side		
		Male (*n* = 148)	Female (*n* = 100)	*p*-Value	Right (*n* = 124)	Left (*n* = 124)	*p*-Value
Spheno-parietal	154 (62.1%)	90 (60.8%)	64 (64%)	0.610	73 (58.9%)	81 (65.3%)	0.294
Fronto-temporal	13 (5.2%)	4 (2.7%)	9 (9%)	0.029 *	7 (5.6%)	6 (4.8%)	0.285
Stellate	3 (1.2%)	2 (1.4%)	1 (1%)	0.802	2 (1.6%)	1 (0.8%)	0.562
Epipteric	29 (11.7%)	6 (4.1%)	26 * (26%)	<0.001 *	19 (15.3%)	13 * (10.5%)	0.254
Synostotic	46 (18.5%)	46 (31.1%)	0 (0%)	<0.001 *	23 (18.5%)	23 (18.5%)	1.000
Total	248 (100%)	148 (100%)	100 (100%)	-	124 (100%)	124 (100%)	-

Asterisk (*) indicates statistically significant difference.

**Table 2 medicina-57-01282-t002:** Bilaterality of the types of pterion and their sex distribution.

Side and Types		Total (*n* = 124)	Sex		*p*-Value
Right	Left		Male (*n* = 74)	Female (*n =* 50)	
Sp	Sp	67 (54%)	41 (54.4%)	26 (52%)	0.711
E	E	9 (7.3%)	1 (1.4%)	7 (14%)	0.005 *
FT	FT	5 (4%)	2 (2.7%)	3 (6%)	0.358
E	Sp	9 (7.3%)	3 (4.1%)	7 (14%)	0.045
SP	E	4 (3.2%)	1 (1.4%)	3 (6%)	0.150
S	Sp	3 (2.4%)	3 (4.1%)	0 (0%)	0.150
SP	S	1 (0.8%)	0 (0%)	1 (2%)	0.225
Sp	FT	1 (0.8%)	0 (0%)	1 (2%)	0.225
FT	Sp	1 (0.8%)	0 (0%)	1 (2%)	0.225
FT	E	1 (0.8%)	0 (0%)	1 (2%)	0.225
Synostotic		23 (18.5%)	23 (31.1%)	0 (0%)	<0.001 *
Symmetric		103 (83.1%)	67 (90.5%)	36 (72%)	-
Asymmetric		18 (16.9%)	7 (9.5%)	14 (28%)	-
Total		124 (100%)	74 (100%)	50 (100%)	-

Asterisk (*) indicates statistically significant difference.

**Table 3 medicina-57-01282-t003:** Mean and standard deviations of distance between center of the pterion to various important bony landmarks by sex and side.

Distance	Total	Mean ± SD (mm)					
		Sex			Side		
		Male	Female	*p*-Value	Right	Left	*p*-Value
PSFZ	35.2 ± 4.9	35.7 ± 4.5	35.0 ± 5.1	0.019	35.8 ± 5.0	34.5 ± 4.8	0.002 *
PZAN	39.5 ± 4.9	39.4 ± 3.8	39.5 ± 5.2	0.127	40.5 + 5.0	38.5 ± 4.6	<0.001 *
PZA	38.9 ± 4.1	38.9 ± 3.4	38.9 ± 4.3	0.088	39.7 ± 4.3	38.0 ± 3.8	<0.001 *
PMP	84.8 ± 4.8	86.4 ± 3.5	84.3 ± 5.1	<0.001 *	85.4 ± 5.1	84.2 ± 4.5	<0.001 *
PH	60.8 ± 3.9	61.4 ± 3.1	60.6 ± 4.1	0.073	60.9 ± 4.5	60.8 ± 3.2	0.820
PI	131.1 ± 5.4	133.8 ± 4.9	130.2 ± 5.3	<0.001 *	130.7 ± 5.5	131.6 ± 5.3	0.001 *
H-width	10.6 ± 4.6	10.2 ± 4.0	10.7 ± 4.8	0.429	10.4 ± 4.7	10.7 ± 4.6	0.5

Asterisk (*) indicates statistically significant difference.

## Data Availability

The data presented in this study are available on request from the corresponding author. The data are not publicly available due to ethical consideration.

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
