# Peer review of "Classification and Morphometric Features of Pterion in Thai Population with Potential Sex Prediction"

_medicina, 2021, doi:10.3390/medicina57111282_

Round 1

Reviewer 1 Report

Paper is very interesting and gives as information in Thai population.  Age has always been of great interest to everyone especially for forensic expert and anthropologists.

The study consisted of 124 skulls, and I think that it would be better that you divide samples by age into 3 groups: the second period of maturation (36-60 years), early aging (61-75 years) and late aging (76-90 years). You could compare these three groups by gender and age in Thai population.

The small errors are yellow highlighted.

Author Response

Dear reviewer,

We would like to thank you for your careful evaluation of our work. Changes suggested by you have now been reflected in this revised version. Please find our point-by-point replies below. Changes made in the manuscript are highlighted in yellow. 

We hope that the manuscript is now acceptable for publication.

Kind regards,

The authors

- The study consisted of 124 skulls, and I think that it would be better that you divide samples by age into 3 groups: the second period of maturation (36-60 years), early aging (61-75 years) and late aging (76-90 years). You could compare these three groups by gender and age in Thai population.
Authors’ reply: We have performed age group-specific analysis according to the reviewer’s suggestion and we found that L_PSZF, L_PZAN, and R_PSFZ measurements were statistically different among the three age groups (using one-way ANOVA) with p-values of <0.001, 0.05 and 0.03, respectively. We have decided not to include this analysis into the results because these findings are already reflected correlational analysis in Figure 4.

- The small errors are yellow highlighted.
Authors’ reply: The errors, especially the surnames in reference number 9 (Knezi et al.), have been corrected.

Reviewer 2 Report

Brief summary 

This study aimed to examine and classify pterion variations and determine the influence of age and gender on the measurements of specific anthropological landmarks, using machine learning algorithms.

General concept comments

Article:

The study is generally well carried out. The use of machine learning algorithms is new and adds a flair to this field of morphometry. However, a clearer, more detailed description of this analysis needs to be done. The Yarusakpong et al, 2021 paper referenced gave a better explanation of what was done. I think this needs to be done here, to improve understanding of the process.

Specific comments: 

Line 69-70: I think it should be the influence of gender and age on the morphometric measurements, not the other way round. This should be applied to all areas this is mentioned including line 228.

Line 183 – I’m not sure if the authors actually mean ‘fetal haemorrhage’ or ‘fatal haemorrhage’, the latter seemed more appropriate, but please check.

Line190-193: Needs to be recast to make better sense.

Author Response

Dear reviewer,

We would like to thank you for your kind comments careful evaluation of our work. Changes suggested by you have now been reflected in this revised version. Please find our point-by-point replies below. Changes made in the manuscript are highlighted in yellow. 

We hope that the manuscript is now acceptable for publication.

Kind regards,

The authors

- The use of machine learning algorithms is new and adds a flair to this field of morphometry. However, a clearer, more detailed description of this analysis needs to be done. The Yarusakpong et al, 2021 paper referenced gave a better explanation of what was done. I think this needs to be done here, to improve understanding of the process.

Authors’ reply: We further added the explanations of the two algorithms similar to what was written in Yurasakpong et al. (2021).

- Line 69-70: I think it should be the influence of gender and age on the morphometric measurements, not the other way round. This should be applied to all areas this is mentioned including line 228.
Authors’ reply: Thank you for pointing this out. This mistake has been rectified.

- Line 183 – I’m not sure if the authors actually mean ‘fetal haemorrhage’ or ‘fatal haemorrhage’, the latter seemed more appropriate, but please check.

Authors’ reply: We have corrected the mistake. The correct word is “fatal”. Thank you.

- Line190-193: Needs to be recast to make better sense.
Authors’ reply: The sentences have been rephrased to improve flow and clarity.

Reviewer 3 Report

The authors examined 124 dry bone to clarify the morphological data of the pterion in Thai population. The theme seems to be a well-known in the anatomical reports, however, they have several unique contents which is considered for publication.

I have read with a great interest and noticed several points to fix.

Abstract

“The surgical landmark for surgical approaches…” is busy sentence. It should be “the landmark for surgical approaches or neurosurgical approaches”.

Introduction

“Pterion is the weakest spot of the skull and can be…” should be divided into two or more sentences. 1st is about the thickness of the pterion and others is mentioned another sentence.

Figure legend in Figure 2

The authors should mention about figure 2F. Figure legend in 2F is missing.

The representative photo of the synostotic type skull (more magnification focused on the pterion) is necessary for better understanding of the readers.

Results

3.3 Randome Rorest classifier was employed for gender and ?

In the previous anatomical study, the size of the atlas correlated well with their height and weight (Yamahata et al. Neurol Med Chir 57:461-466, 2017). I think the length between the center of the pterion and other locations depends on the original height or weight of the cadaver head to some extent. If the authors obtain the data, statistical analysis with regard to the height and weight will give another important data.

Unfortunately, the variation of the pterion makes no difference for the operation in the clinical field of neurosurgery. The distance between the center of the pterion and others also has no impact on the operative procedures. However, as the authors mentioned in the content, it may have important meaning for the legal medicine, human identification, and anthropology. The thickness around the pterion is more important for the neurosurgical operations. I request the authors to examine the relationship between the classification of the pterion and the thickness around the pterion in the future study.

Author Response

Dear reviewer,

We would like to thank you for your careful evaluation of our work. Changes suggested by you have now been reflected in this revised version. Please find our point-by-point replies below. Changes made in the manuscript are highlighted in yellow. 

We hope that the manuscript is now acceptable for publication.

Kind regards,

The authors

- Abstract: “The surgical landmark for surgical approaches…” is busy sentence. It should be “the landmark for surgical approaches or neurosurgical approaches”

Authors’ reply: The sentence has been rewritten. Thank you!

- Introduction: “Pterion is the weakest spot of the skull and can be…” should be divided into two or more sentences. 1st is about the thickness of the pterion and others is mentioned another sentence.
Authors’ reply: The sentence has been divided. Thank you!

- Figure legend in Figure 2: The authors should mention about figure 2F. Figure legend in 2F is missing.

Authors’ reply: Figure legend has been corrected to include Figure 2F.

- The representative photo of the synostotic type skull (more magnification focused on the pterion) is necessary for better understanding of the readers.
Authors’ reply: Magnification has been added to figure 2F.

- Randome Rorest classifier was employed for gender and ?
Authors’ reply: The word “and” has been replaced with “prediction”.

In the previous anatomical study, the size of the atlas correlated well with their height and weight (Yamahata et al. Neurol Med Chir 57:461-466, 2017). I think the length between the center of the pterion and other locations depends on the original height or weight of the cadaver head to some extent. If the authors obtain the data, statistical analysis with regard to the height and weight will give another important data.
Authors’ reply: Unfortunately, height and weight of the donors were unknown. This is now noted in the last paragraph of the discussion as a limitation.

- Unfortunately, the variation of the pterion makes no difference for the operation in the clinical field of neurosurgery. The distance between the center of the pterion and others also has no impact on the operative procedures. However, as the authors mentioned in the content, it may have important meaning for the legal medicine, human identification, and anthropology. The thickness around the pterion is more important for the neurosurgical operations. I request the authors to examine the relationship between the classification of the pterion and the thickness around the pterion in the future study.
Authors’ reply: We acknowledge the limited application of our findings in neurosurgery. We agree that the relationship between thickness and classification could be our next study. Thank you for your suggestion!!